# Coincidence Detection Is All You Need

## Abstract

This paper demonstrates that the performance of coincidence detection - a classic
neuromorphic signal processing method found in Rosenblatt's perceptrons with
distributed transmission times, can be competitive to a state-of-the-art deep learning
method for pattern recognition. Hence, we cannot remain comfortably numb to the
prevailing dogma that efficient matrix-vector operations is all we need; but should
enquire with greater vigour if more advanced continual learning methods (running
on spiking neural network hardware with neuromodulatory mechanisms at multiple
timescales) can beat the accuracy of task-specific deep learning methods.

## 1   Introduction

Frank Rosenblatt and his team (1957-1971) built and analyzed several kinds of perceptrons [1, 2, 3, 4]
- networks of sensory, association and receptor neurons; which in contemporary deep learning termi-
nology relates to the input, hidden and output layers. The propagating signals were binary (compatible
with a spike-based view), the synaptic delays (transmission times) and weights (memory states) could
be analog, the network could be recurrent and was often randomly interconnected, and learning
often meant tuning the weights of the association-receptor subnetwork by some error-corrective
reinforcement. The synaptic delays were not learnt but instead randomly distributed in Rosenblatt's
Tobermory perceptrons [5], and this was rich enough to realize concentration-invariant and uniform
time-warp invariant spatiotemporal classification by logarithmic encoding and coincidence detection.
However, the processing speed of commercial Von Neumann computers advanced exponentially
and outperformed neuromorphic hardware on yesterdecade's benchmarks [6]. The Tobermory per-
ceptron was forgotten, nevertheless, the utility of logarithmic encoding and coincidence detection
was formalized by John Hopfield [7] as an efficient solution to the *analog match* problem in pattern
recognition.

Now, half a century after the accidental demise of Rosenblatt, neuromorphic signal processors are
making a comeback. For example, (1) Intel's Loihi with spike-time dependent plasticity mechanisms
for learning olfactory pattern recognizers [8]; (2) Physical reservoir computing networks [9] where
the interconnectivity of the hidden layer is unchanged, closer to the spirit of Rosenblatt's randomly
interconnected sensory-association subnetwork.

Here, to strengthen the case for revisiting classic methods on novel and modern hardware, we evaluate
the performance of coincidence detection in comparison to a deep learning method. Nothing more,
nothing less, although this work was triggered by a rabid interest in employing artificial intelligence
to sniff out infections and prevent future pandemics.

Submitted to 36th Conference on Neural Information Processing Systems (NeurIPS 2022). Do not distribute.

Table 1: Test accuracy (%)

| ResNet-26 | Coincidence detection |
| --- | --- |
| 82.2± 0.3 (from [10]) | 82.7 (this work) |

## 2 Methods

Here, we consider the work [10] of an interdisciplinary team, where a 26 layer convolutional neural network with residual connections (ResNet-26) was successfully trained for classifying pathogenic bacteria by Raman spectroscopy. In their work, there are $N = 30$ classes of bacterial isolates and they begin with a ResNet-26 pre-trained on $N \times 2000$ spectra, then for each class $n = 1 : N$ there are $M = 100$ training spectra, and similarly $N \times M = 3000$ test spectra. Each spectrum $x$ contains 1000 floating-point numbers ranging between 0 and 1. Although compute intensive, their deep learning method proved to be a tool of great convenience for pattern recognition in a challenging dataset, where intra-isolate spectra were often more dissimilar than inter-isolate spectra.

Our method to tackle the above dataset, is inspired by the theory of how coincidence detection [7] in animal brains is fundamental for odour classification in complex and turbulent mixtures. Each class $n$ has a vector representation $w_n$ that is learnt, and an input vector $x$ results in an output class $y(x) = \arg_n \max(x \bigwedge w_n)$ where we introduce the operator $\bigwedge$ to represent the coincidence between two signals. The analytical nature of coincidence detection depends on the specificities of the ion-channels and the membranes involved [11], and may even incorporate nonlinear leaky-integrate [12] multiple timescale mechanisms. We do not yet have a complete theory of neuromorphic signal processing, so here we introduce an approximation for the translation and scale-invariant property of coincidence detection as

$$\arg_n \max(x \bigwedge w_n) \approx \arg_n \max(w_n \cdot \hat{x}), \tag{1}$$

where $\hat{x}$ is the zero-mean unit-variance normalization of $x$.

Thus, the approximation in Eq. (1) allows $y(x)$ to be learnt by a logistic regression on the normalized dataset. We discard the pre-training data, pre-process the training and test spectra by a range-1 mean filter, and use the default method for logistic regression in Wolfram Mathematica (L2-regularization = 0.0001, optimization method = limited-memory BFGS). Code is provided in the supplemental material for reproducibility.

## 3 Result and outlook

The coincidence detection (via normalized logistic regression) method introduced here achieves a test accuracy greater than ResNet-26 (see Table 1), and it took less than 3 seconds to train the classifier on a modern desktop (without any special-purpose GPUs). Check the Appendix for a confusion matrix plot of the training and test data. Note that the training data was fit all at once to a 100% accuracy. With a more neuromorphic coincidence detection method and a learning method that adapts the synaptic delays $w$ continually, to keep track under changing environmental conditions, we may achieve even greater accuracies.

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

# A   Appendix

Confusion matrix of the training and test data. Wolfram Mathematica code to reproduce these results is provided as supplemental material.

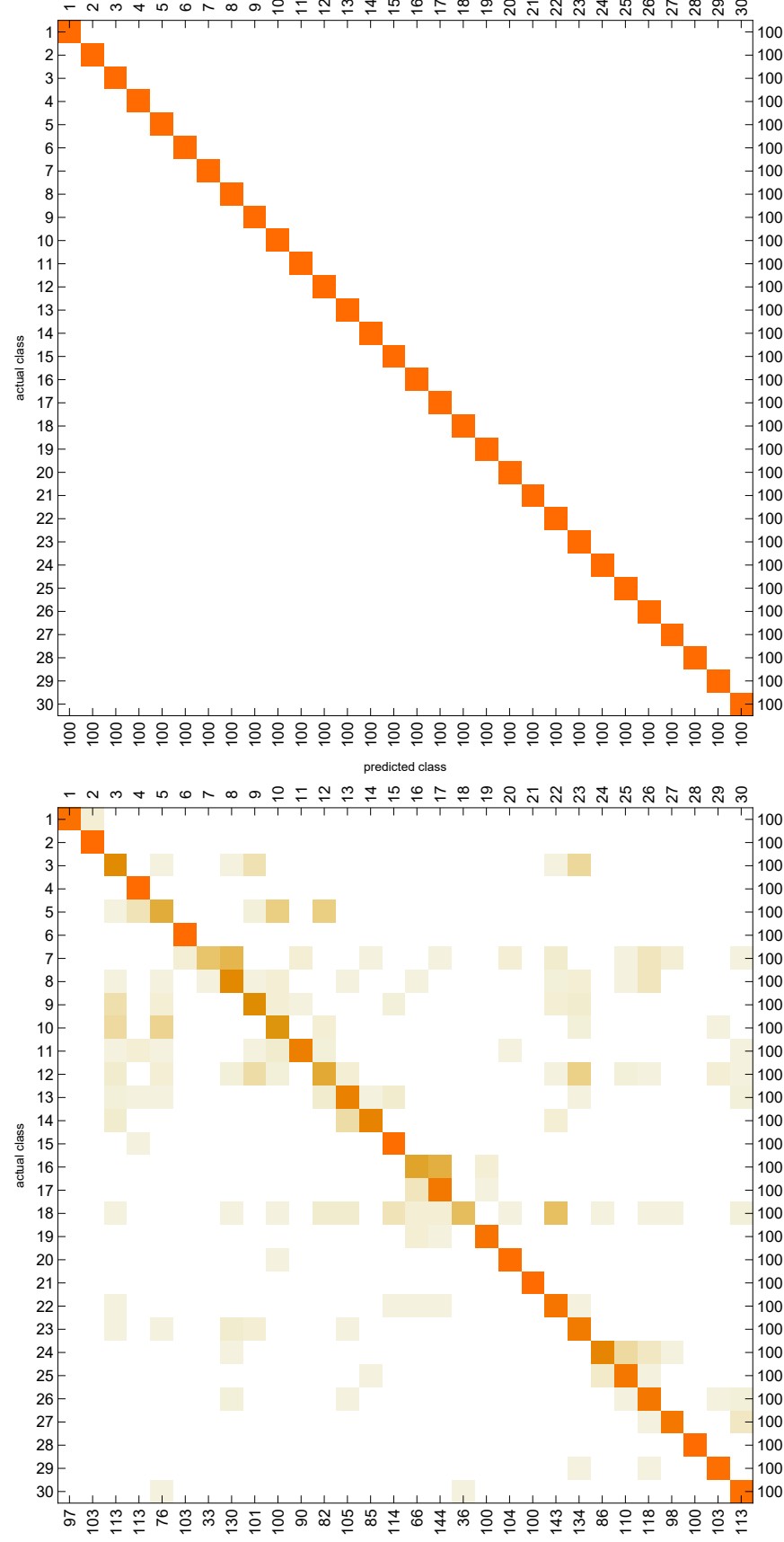

