# Classification by coincidence detection via Sample-Standardized Softmax regression (S3-classifier)

Here we **introduce the S3-classifier**, which was **inspired by the theory of coincidence detection**, and show it to be **competitive to ResNet-26** based on data from

> Ho, CS. *et al*. Rapid identification of pathogenic bacteria using Raman spectroscopy and deep learning. *Nat Commun* 10, 4927 (2019). https://doi.org/10.1038/s41467-019-12898-9

used under http://creativecommons.org/licenses/by/4.0/.

The **main results were first obtained using Wolfram Mathematica** by importing the dataset as matrices using the ReadNumpy package by Luca Robbiano, and running the following analysis

```
# Wolfram Language code to be run on the imported matrices xTrain and xTest
ssxTrain = Table[ Standardize@MeanFilter[xTrain[[L]], 1], {L, 3000}];
trainingset = Catenate@Table[ssxTrain[[(i - 1)*100 + j]] -> i, {i, 30}, {j, 100}];
c = Classify[trainingset, Method -> "LogisticRegression", PerformanceGoal -> "DirectTraining"];
ssxTest = Table[ Standardize@MeanFilter[xTest[[L]], 1] , {L, 3000}];
testset = Catenate@Table[ssxTest[[(i - 1)*100 + j]] -> i, {i, 30}, {j, 100}];
ClassifierMeasurements[c, testset]
```

**Python code is provided below for a more detailed analysis** , as it is presently more popular among the machine learning community.

In [ ]:
```
from time import time
t00 = time()
import numpy as np
from sklearn.preprocessing import scale
from sklearn.linear_model import LogisticRegression,LogisticRegressionCV
import matplotlib.pyplot as plt
%matplotlib inline
```

Visualize the train and test dataset

In [ ]:
```
X_train = np.load('./drive/MyDrive/X_finetune.npy') #3000 samples of Raman spectra with 1000 lines each
y_train = np.load('./drive/MyDrive/y_finetune.npy') #3000 samples of bacterial classes from 0-29
X_test = np.load( './drive/MyDrive/X_test.npy')
y_test = np.load('./drive/MyDrive/y_test.npy')
fig, (ax1,ax2) = plt.subplots(1,2,figsize=(30,50))
ax1.matshow(X_train)
ax2.matshow(X_test)
plt.show()
```

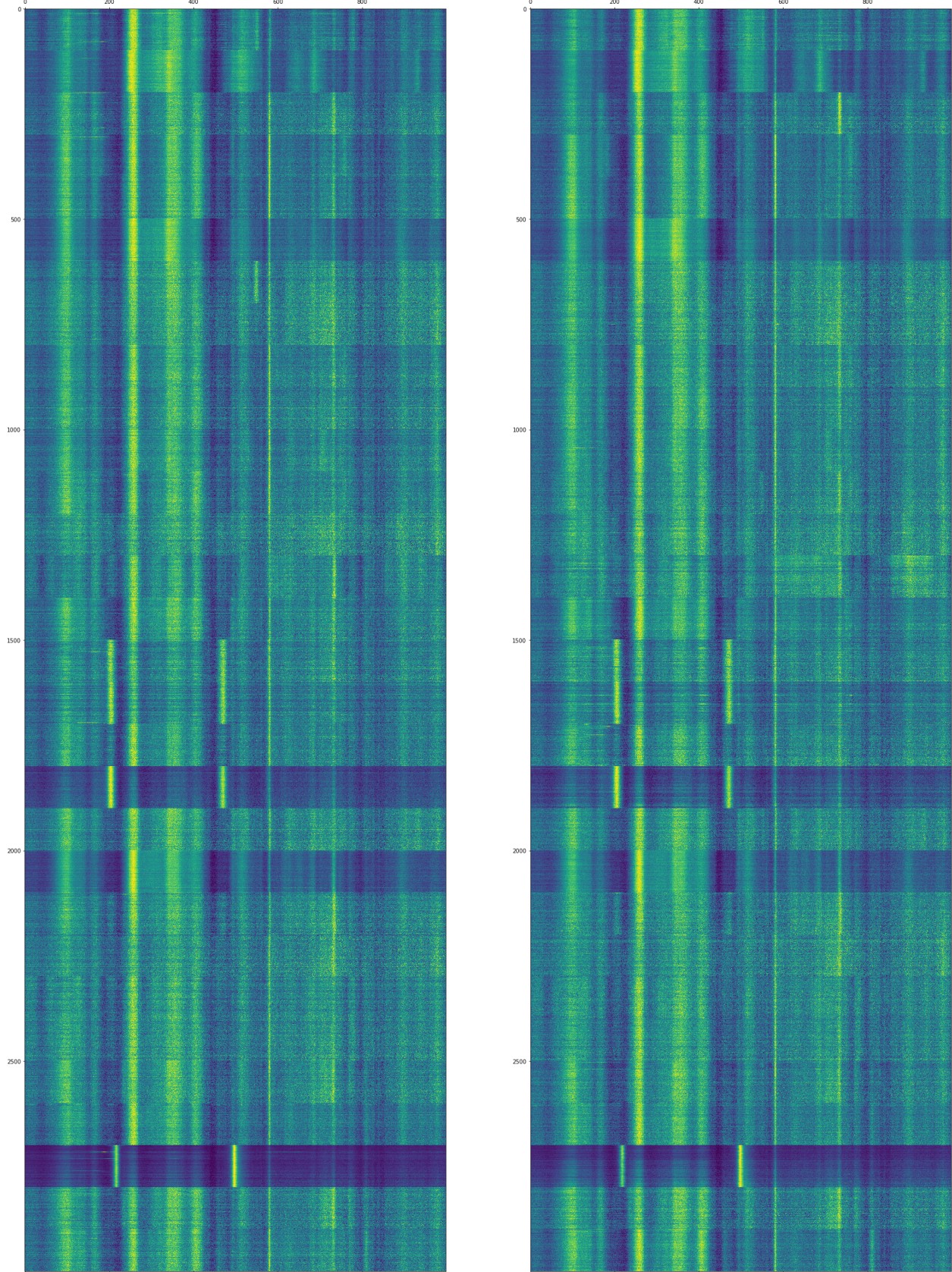

Fit a softmax (multinomial logistic) regression model to the sample-standardized training dataset

```
In [ ]:  sX_train = scale(X_train,axis=1) #note that standardization is performed across samples instead of across features
         reg = LogisticRegression(penalty='l2',C=1e4,max_iter=100,solver='lbfgs',multi_class='multinomial').fit(sX_train, y_train)
         reg.score(sX_train, y_train)
```

```
/usr/local/lib/python3.7/dist-packages/sklearn/linear_model/_logistic.py:818: ConvergenceWarning: lbfgs failed to converge
(status=1):
STOP: TOTAL NO. of ITERATIONS REACHED LIMIT.

Increase the number of iterations (max_iter) or scale the data as shown in:
    https://scikit-learn.org/stable/modules/preprocessing.html
Please also refer to the documentation for alternative solver options:
    https://scikit-learn.org/stable/modules/linear_model.html#logistic-regression
  extra_warning_msg=_LOGISTIC_SOLVER_CONVERGENCE_MSG,
```

 1.0

Evaluate the model on the standardized and smoothened test dataset

```python
#standardization is performed across samples instead of across features
sX_test = scale(X_test,axis=1)
#smooth dataset to account for horizontal shifts in the line spectra
def smooth(mat):
    return np.hstack((mat[::,0:1]+mat[::,1:2],mat[::,0:-2]+mat[::,1:-1]+mat[::,2:],mat[::,-1:]+mat[::,-2:-1]))
ssX_test = smooth(sX_test)
print(reg.score(ssX_test, y_test))
```

0.8293333333333334

> The accuracy of our **theory-inspired model** is 82.9% and **outperforms** the 82.2% obtained by the ResNet-26 **deep learning** method.

```python
# prepare results to be plotted
y=y_test
y_hat=reg.predict(ssX_test)
```

---

The remaining analysis employs the code by Chi-Sing Ho to produce confusion matrix plots of the same style as in Ho, CS. *et al* (2019).

## Plotting confusion matrix for bacterial isolates

We use the predictions to plot a version of the confusion matrix see in Figure 2 of the paper (Ho *et al*. 2019). Each row represents the true class and each columen represents the predicted class. The entries of the confusion matrix are normalized so that the rows sum to 100% (differences from rounding). The accuracy for each class can be seen in the diagonal entries.

```python
import seaborn as sns
from sklearn.metrics import confusion_matrix

ORDER = [16, 17, 14, 18, 15, 20, 21, 24, 23, 26, 27, 28, 29, 25, 6, 7, 5, 3, 4,
         9, 10, 2, 8, 11, 22, 19, 12, 13, 0, 1]

STRAINS = {}
STRAINS[0] = "C. albicans"
STRAINS[1] = "C. glabrata"
STRAINS[2] = "K. aerogenes"
STRAINS[3] = "E. coli 1"
STRAINS[4] = "E. coli 2"
STRAINS[5] = "E. faecium"
STRAINS[6] = "E. faecalis 1"
STRAINS[7] = "E. faecalis 2"
STRAINS[8] = "E. cloacae"
STRAINS[9] = "K. pneumoniae 1"
STRAINS[10] = "K. pneumoniae 2"
STRAINS[11] = "P. mirabilis"
STRAINS[12] = "P. aeruginosa 1"
STRAINS[13] = "P. aeruginosa 2"
STRAINS[14] = "MSSA 1"
STRAINS[15] = "MSSA 3"
STRAINS[16] = "MRSA 1 (isogenic)"
STRAINS[17] = "MRSA 2"
STRAINS[18] = "MSSA 2"
STRAINS[19] = "S. enterica"
STRAINS[20] = "S. epidermidis"
STRAINS[21] = "S. lugdunensis"
STRAINS[22] = "S. marcescens"
STRAINS[23] = "S. pneumoniae 2"
STRAINS[24] = "S. pneumoniae 1"
STRAINS[25] = "S. sanguinis"
STRAINS[26] = "Group A Strep."
STRAINS[27] = "Group B Strep."
STRAINS[28] = "Group C Strep."
STRAINS[29] = "Group G Strep."
```

```python
# Plot confusion matrix
sns.set_context("talk", rc={"font":"Helvetica", "font.size":12})
label = [STRAINS[i] for i in ORDER]
cm = confusion_matrix(y, y_hat, labels=ORDER)
plt.figure(figsize=(15, 12))
cm = 100 * cm / cm.sum(axis=1)[:,np.newaxis]
ax = sns.heatmap(cm, annot=True, cmap='YlGnBu', fmt='0.0f',
                 xticklabels=label, yticklabels=label)
ax.xaxis.tick_top()
plt.xticks(rotation=90)
plt.show()
```

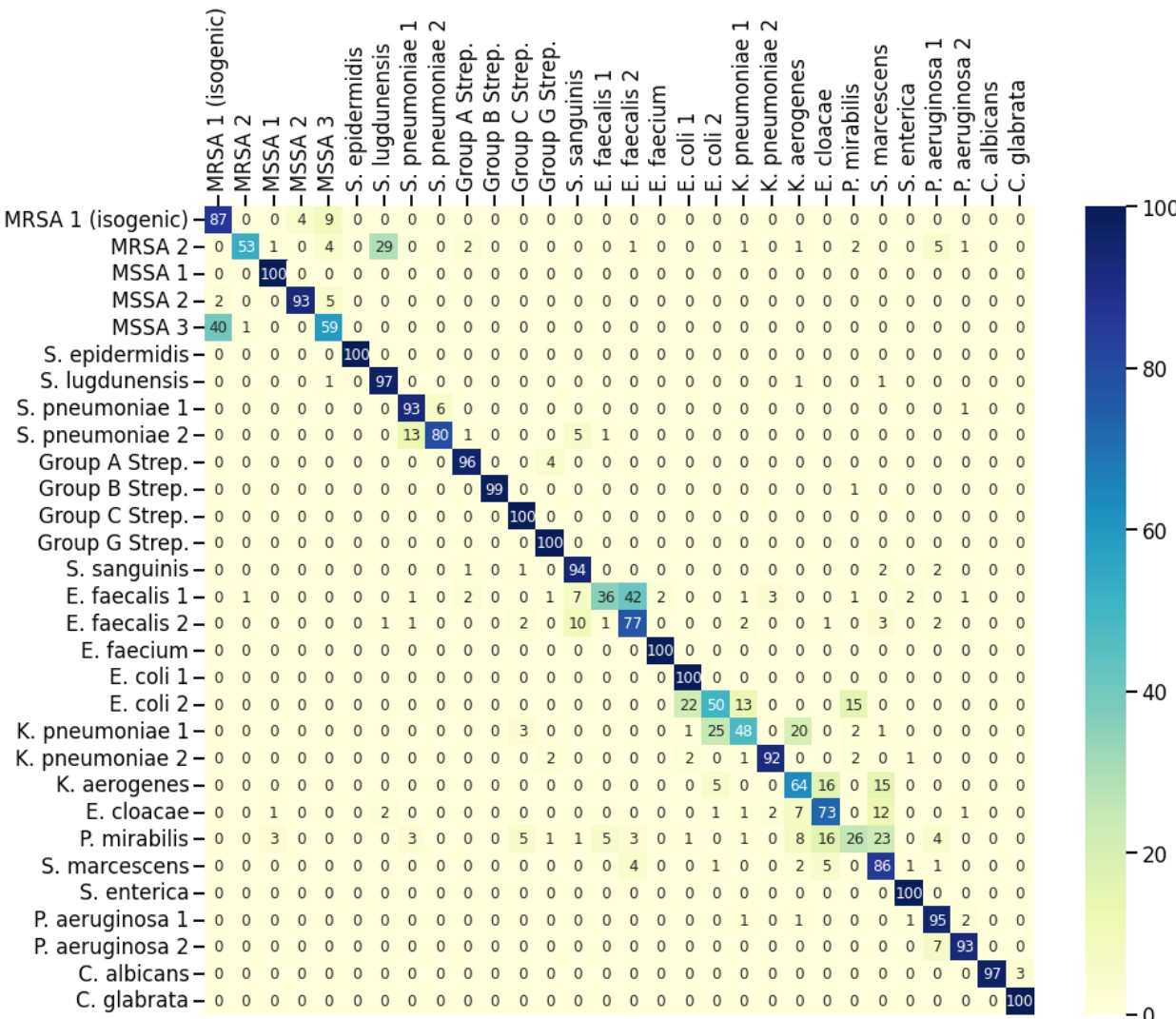

## Plotting confusion matrix for antibiotic groupings

Finally, we can combine predictions into antibiotic groupings to estimate treatment accuracy. The entries of the confusion matrix are normalized so that the rows sum to 100% (differences from rounding). The accuracy for each antibiotic group can be seen in the diagonal entries.

```
In [ ]:   ATCC_GROUPINGS = {3: 0,
                           4: 0,
                           9: 0,
                           10: 0,
                           2: 0,
                           8: 0,
                           11: 0,
                           22: 0,
                           12: 2,
                           13: 2,
                           14: 3,  # MSSA
                           18: 3,  # MSSA
                           15: 3,  # MSSA
                           20: 3,
                           21: 3,
                           16: 3,  # isogenic MRSA
                           17: 3,  # MRSA
                           23: 4,
                           24: 4,
                           26: 5,
                           27: 5,
                           28: 5,
                           29: 5,
                           25: 5,
                           6: 5,
                           7: 5,
                           5: 6,
                           19: 1,
                           0: 7,
                           1: 7}
          ab_order = [3, 4, 5, 6, 0, 1, 2, 7]
          antibiotics = {}
          antibiotics[0] = "Meropenem" # E. coli
```

```
antibiotics[1] = "Ciprofloxacin" # Salmonella
antibiotics[2] = "TZP" # PSA
antibiotics[3] = "Vancomycin" # Staph
antibiotics[4] = "Ceftriaxone" # Strep pneumo
antibiotics[5] = "Penicillin" # Strep + E. faecalis
antibiotics[6] = "Daptomycin" # E. faecium
antibiotics[7] = "Caspofungin" # Candidas
```

In [ ]:
```
# Mapping predictions into antibiotic groupings
y_ab = np.asarray([ATCC_GROUPINGS[i] for i in y])
y_ab_hat = np.asarray([ATCC_GROUPINGS[i] for i in y_hat])
# Computing accuracy
acc = (y_ab_hat == y_ab).mean()
print('Accuracy: {:0.1f}%'.format(100*acc))
```

Accuracy: 96.9%

In [ ]:
```
sns.set_context("talk", rc={"font":"Helvetica", "font.size":12})
label = [antibiotics[i] for i in ab_order]
cm = confusion_matrix(y_ab, y_ab_hat, labels=ab_order)
plt.figure(figsize=(5, 4))
cm = 100 * cm / cm.sum(axis=1)[:,np.newaxis]
ax = sns.heatmap(cm, annot=True, cmap='YlGnBu', fmt='0.0f',
                 xticklabels=label, yticklabels=label)
ax.xaxis.tick_top()
plt.xticks(rotation=90)
plt.show()
print('\n This demo was completed in: {:0.2f}s'.format(time()-t00))
```

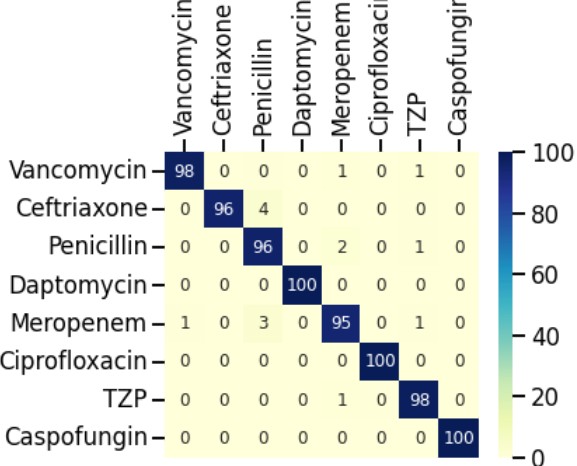

This demo was completed in: 27.12s