# OpenReview forum: "Coincidence Detection Is All You Need"
_NeurIPS.cc/2022/Conference — NeurIPS 2022 Submitted_

### Official Review · Reviewer_QphW · 2022-07-07

**Rating:** 2
**Confidence:** 4
**Soundness:** 1 poor
**Presentation:** 1 poor
**Contribution:** 2 fair

**Summary:**

The authors make an argument for revisiting the use of neural circuits for pattern recognition problems citing the existence of similar neural circuits in animal brains. In that direction, the authors refer to the experiments in [10], which uses deep learning to classify pathogenic bacteria.
With motivation from coincidence detection theory, the authors propose a multinomial logistic regression model to solve the problem. The dataset is preprocessed with a specific standardization and smoothening that allows the model to demonstrate the translation and scale-invariance of coincidence detection. The results show higher accuracy than the residual neural network used in [10] at isolate-level classification.


**Questions:**

Authors should consider generating more stats on their accuracy % and provide a more thorough comparison with the baseline (ResNet-26).
Further, authors should share additional experiments breaking down the contribution of standardization and smoothing steps. Lastly, explaining why their model fares better than the deep learning model for this problem will be helpful to understand the kind of problems this model can help solve.


**Limitations:**

The authors do not discuss the limitations of their work, kindly refer to Questions section for suggestions on how to improve this work.

**Strengths And Weaknesses:**

The authors show a higher accuracy compared to a deep learning algorithm which helps show the potential of framing classification problems with the coincidence detection approach. However, the authors show the accuracy for one single train/validation split whereas the reference [10] shows a statistical mean of 82.2 +- 0.3% over 5 runs. For a fair comparison the authors should rerun the experiment and generate statistical mean/variance results for random train/validation split to remove any bias due to specific train/validation split selection. Additional details about the baseline model (ResNet-26) e.g., network architecture/training time/training accuracy would be helpful to get an all-round comparison.

Further, the authors provide no explanation as to why their model fares better than the deep learning model for the specific problem being addressed. The paper lacks any detailed analysis on the contribution of the standardization and smoothening steps which could be generated by selectively applying these preprocessing steps.

Lastly, the authors have added a few figures to their supplementary material which would have been valuable additions to the main paper. A discussion on these figures to help understand their results and how it compares to the baseline results would be helpful. For instance, the baseline has a higher antibiotic treatment classification accuracy than the proposed model, but this isn’t discussed and no justification is provided for the same.

---

> ### Author Response · Authors · 2022-07-26
> **The focus is on Coincidence detection for solving the analog match problem- perhaps the first empirical demonstration on a real world dataset**
>
> The reviewer asks for more stats, but is it not futile? Given that this is anyhow based on performance on a single dataset?
> The focus of this paper is to demonstrate that the approximation for coincidence detection introduced here is able to solve an analog match problem (discussed insightfully by Hopfield [7], but not as well-known as it should be). That the model fares slightly better is a bonus, actually deep learning methods can surely learn a coincidence detector (albeit in a computationally expensive way).
>
> Moreover, in order to ensure reproducibility, the method was tested in two programming languages Mathematica (yielding an accuracy of 82.7 % as reported in the main text) and Python (yielding an accuracy of 82.9% as reported in the supplementary material).

---

> > ### Comment · Reviewer_QphW · 2022-08-09
> > **Empirical demonstration while helpful has limited impact**
> >
> > Adding stats would ideally only strengthen the presented results and verify against strong variation in accuracy due to different split of train/test data. I agree that good accuracy across multiple language environments adds some support against the high variation in accuracy. However, without added stats, the result while impressive has limited impact (for the given dataset and train/test splits).
> >
> > As mentioned previously, testing coincidence detection for other similar analog match problems might help build the case for proposed ideas. Further, that might also help identify dataset features that might help identify when to use coincidence detection to get the high accuracy results similar to what you observed.

---

> > > ### Author Response · Authors · 2022-08-09
> > > **Testing coincidence detection for other datasets is welcome**
> > >
> > > Yes, we need more datasets for future work. But it is not so wise to demand that for  "first" empirical  demonstrations, if you want to accelerate impact coming from outside the resource-rich industry.

---

### Official Review · Reviewer_V6Wx · 2022-07-11

**Rating:** 1
**Confidence:** 4
**Soundness:** 1 poor
**Presentation:** 2 fair
**Contribution:** 1 poor

**Summary:**

The authors draw inspiration from Rosenblatt’s perceptrons to investigate if coincidence detection can be an important element for neuromorphic signal processing. To illustrate their approach, the authors apply a perceptron to a bacteria identification problem, showing that it outperforms a recent deep neural network approach.

**Questions:**

1. Is there no more suitable implementation of coincidence detection, e.g., within a spiking net?
2. Is your model in eq 1 not simply a perceptron? (With normalized inputs and a max on the outputs)
3. Can you report on other methods on the task of ref [10]?
4. Can you show similarly good performance on other tasks?

**Strengths And Weaknesses:**

Strengths:
* The paper is thought-provoking, drawing attention to how early works on neural networks focused on neural temporal dynamics that are now making a come-back in the SNN domain.
* Coincidence detection as a basis for analog pattern recognition (as explained by Hopfield, [7] in the paper) is highly interesting.

Weaknesses:
* Arriving at equation 1 was a desillusion for me, as I very much liked the text leading to it. In this equation, coincidence detection is approximated with a linear multiplication (and a max function for picking the most probable class). This takes away the main idea of coincidence detection, which requires temporal delays.
* The simple "coincidence" detector gives very good results compared with a deep net. Although this could be demonstrating an advantage of coincidence detection, it may also be that the classification problem is actually not that difficult. Paper [10] seems to only apply a deep net to the problem. The authors only apply a linear function. What do other functions do? k-nearest  neighbors, SVMs, ...?
* The paper is very short and fails to delve into the details of why the proposed method works so well. It seems logical that it has to be applied to other tasks as well to get more insight.

---

> ### Author Response · Authors · 2022-07-26
> **More than trivial, it is profound simplicity!**
>
> 1. Yes, references [11] and [12] point to this, but are expensive to implement on conventional hardware. Future work should compare how the approximate implementation of coincidence detection compares to more advanced methods on neuromorphic hardware.
>
> 2. Yes, is it not beautiful? Did you notice that the normalization is performed across a different axis in comparison to the standard suggestion of Python sklearn for logistic regression?
>
> 3. No, because this is not the focus of the paper. The message is, here is a novel method found by theory with better performance in comparison to the impactful deep learning method by a large team of researchers in Stanford university, cited over 250 times.
>
> 4. This is an important question and is an avenue for future work. Word needs to get out, so that more researchers are inspired by [7] and this work.
>
> Hope these answers provide you with a greater appreciation of this work. Although this paper is very short, it needs rumination to get the best out of it.

---

> > ### Comment · Reviewer_V6Wx · 2022-08-07
> > **Response**
> >
> > Again, although I am sympathetic to the authors' efforts, I do not think the current paper is insightful enough for publication at NeurIPS.
> >
> > Re 1/2. Although a traditional perceptron is indeed beautiful, it is not exactly novel. What piqued my interest was that the original studies actually went beyond the final ANN perceptron model, focusing on coincidence detection and temporal dynamics. To me, studying these aspects should not be left to future work.
> >
> > Re 3. I do think this point is relevant for the current paper. Is it your approximation to coincidence detection that leads to these good results or is it simply that the studied task is easier than was previously thought? Nowadays, deep neural networks are the go-to method for tackling almost any task, so it does not surprise me that this is used in ref [10]. They could (or should?) have tried also other methods. But the authors of this article definitely should study other, traditional methods to piece apart whether the task is simpler than thought before, or whether it is really the "coincidence detection" of the perceptron that makes the difference (although I would not call it coincidence detection, since there are no delays).
> >
> > Re 4. I think multiple tasks are essential to show that the result is not just a fortunate result.
> >
> > I encourage the authors to continue their line of work, but remain at my previous evaluation.

---

> > > ### Author Response · Authors · 2022-08-07
> > > **Need better understanding not sympathy!**
> > >
> > > 2. This is not a perceptron with a conventional learning method. Again, did you notice that the normalization is performed across a different axis in comparison to the standard suggestion of Python sklearn for logistic regression?
> > >
> > > 3. Ref. [10] already explored traditional methods (k-NN, SVM) and justified their choice for a deep learning method. It would not be cited over 250 times otherwise. It is true that this work is an approximation of coincidence detection, and hence is meant to be a teaser paper to attract attention towards truly neuromorphic algorithms which oddly is a lost focus among the NeurIPS community (for example, even your primal calls for k-NN and SVM are unwise- you are focussing on machine learning instead of the science of neural information processing systems). Thus, the focus is here is exclusively on coincidence detection for solving the analog match problem- perhaps the first empirical demonstration on a real world dataset.
> > >
> > > 4. Agreed. Would you have the resources to execute this idea for multiple tasks? Or just anyone who is interested can contact by email (if this paper is rejected and stands still and de-anonymized on OpenReview).

---

> > > > ### Comment · Reviewer_V6Wx · 2022-08-08
> > > > **Response**
> > > >
> > > > Re 2. If the authors would like me to better understand, can they then please explain to me how this is not a perceptron... Eq. shows an argmax and a product of two vectors right? Hence, we have an input and output layer, where each output neuron is a weighted sum of the inputs... The absence of hidden layers would make this a perceptron with a linear activation function to me, followed by an argmax to pick the most active neuron / class.
> > > >
> > > > That the authors normalize the vectors in a different way only changes what goes into the perceptron. The authors ask if it is clear to me that normalization is performed on a different axis. No, this does not become clear in the paper... From the responses to other reviewers, I distill that the normalization is not over the dataset / samples, as not the whole dataset is necessary. I then guess that the normalization is done over the vector of a single sample itself (the features in the feature vector)? This could be a very bad idea for certain datasets in which different features in the feature vector have very different magnitudes. However, in the code in the supplementary material, the comment tells me that the normalization *is* done over the samples. If not over the whole dataset, then over how many samples / how? In conclusion, please explain me / readers how normalization is done (not only here, but in the paper as well). Then, please show the influence of that normalization on the results, also for other tasks / data sets.
> > > >
> > > > Re 3. My apologies. When performing my review, I actually checked whether [10] did this, but I overlooked it. It could be then that the success of the proposed method indeed lies in the normalization. It would be great to see an analysis of that.
> > > >
> > > > The relation suggested between the exact content and number of citations does not really seem causal to me. Studies can be cited for many different reasons. I agree that it is interesting that the current article achieves slightly better results with a much simpler approach.
> > > >
> > > > On the other hand, I disagree with the statement that the NeurIPS community does not give attention to truly neuromorphic work - there is a lot of attention for that at the moment. I would, however, not label the article of the authors as neuromorphic. Again, for real coincidence detection, temporal dynamics are required.

---

> > > > > ### Author Response · Authors · 2022-08-08
> > > > > **Technical acceptance by experts is an incentive to spend resources on a tutorial-style presentation for the wider community**
> > > > >
> > > > > 2. Thank you for spotting an error, indeed the comment in the supplementary information reads wrongly. This will be corrected and a formula for the zero-mean unit-variance normalization will be provided on line 51 in the main-text, if the paper is accepted.
> > > > >
> > > > > Indeed CONVENTIONAL wisdom is that it is a bad idea to do a normalization in this way, which is why perceptrons were not employed with this kind of pre-processing until now. This paper instead argues from the theory of coincidence detection that it is actually a good idea for preprocessing datasets that are compatible with the analog match problem, which turns out to be true upon evaluation in this empirical dataset. Future work must find other empirical real world datasets where this is true (neuromorphic olfaction, to sniff out infections and prevent future pandemics is one such target application).
> > > > >
> > > > > 3. Lines 46 to 48 agree with your interpretation about temporal dynamics being required. That being said, you should also realize that Eq. (1) has a sense of neuromorphic-ness that SVM does not have. Also, the lack of temporal dynamics is again referred to when the paper is concluded as "With a more neuromorphic coincidence detection method and a learning method that adapts the synaptic delays continually, to keep track under changing environmental conditions, we may achieve even greater accuracies.".

---

### Official Review · Reviewer_ctyh · 2022-07-19

**Rating:** 2
**Confidence:** 5
**Soundness:** 1 poor
**Presentation:** 1 poor
**Contribution:** 2 fair

**Summary:**

Authors present an empirical study on exploiting a classical coincidence detection based method for classification. Experiments on classifying pathogenic bacteria from Raman spectroscopy data is performed, showing a 0.5% increase in classification accuracy in comparison to using a deep learning approach with a ResNet-26 as performed in [Ho et al., 2019].

**Questions:**

There is an interesting empirical observation here, yet the narrative is too shallow and clarity of the paper is weak. Generally, it is still not clear in text how the proposed "coincidence detection" application is applied without checking the implementation (lack of a thorough Methods section). There is also no relationship drawn with any existing neuromorphic processing algorithms that exploit temporal information encoding (lack of a thorough Related Work section). Perhaps the authors may consider extending their findings with a clear methodological contribution in a complete manuscript.


**Limitations:**

No discussions regarding potential societal impact or limitations.

**Strengths And Weaknesses:**

Paper is written as a very short report on an empirical observation. No details on the relevance of the approach to existing methods are present. A clear narrative on the methodological contribution is lacking. It is not really suitable for publication at NeurIPS in its current form.

---

> ### Author Response · Authors · 2022-07-26
> **More than an interesting empirical observation- a novel method was found by theory!**
>
> That the paper is written as a very short report on an empirical observation, would be a strength of this paper. The result in table-1 speaks for itself (i.e. here is a novel method with better performance in comparison to the impactful deep learning method by a large team of researchers in Stanford university, cited over 250 times).  Of course, this novel method will need to be applied to other datasets (which is why it needs to be presented in a conference to gain the attention of fellow researchers). Moreover, references [7], [11], [12] have been thoughtfully chosen as related work. Would the reviewer have any suggestions for related work that are distinctly important? Happy to include or debate over them.

---

> > ### Comment · Reviewer_ctyh · 2022-08-08
> > **Thanks for the responses**
> >
> > Thanks to the authors for their responses.
> >
> > I certainly agree on the remark that a condensed writing could be a strength, if the contribution is clear to any reader and self-contained. The paper maintains its previous form, and unfortunately I don't see improvements to the clarity of manuscript with regards to any reviewer comments. I believe this very interesting result could be presented in better shape for a stronger impact at NeurIPS.
> >
> > One example: As mentioned before, how the method is applied (and its contributions) is still not clear in the report (e.g., it now becomes apparent in the code supplement that sample-wise standardization for preprocessing is applied). When one applies this standardization (which appears to be a contribution here) on the whole test set, what do we really assume? Do the existing DL approach in Table 1 [Ho et al] also consider all test samples to be available at inference time (this is the case in the authors' approach since the test set preprocessing statistics are computed over all samples)? Couldn't we discuss such methodological technicalities or highlight them in text?
> >
> > My opinion and score remains the same, and I don't believe the paper is ready for publication at this stage.

---

> > > ### Author Response · Authors · 2022-08-08
> > > **Technical acceptance by experts is an incentive to spend resources on a tutorial-style presentation for the wider community**
> > >
> > > In line 51 of the main paper the sample-standardization $\hat {\boldsymbol{x}}$ of a sample $\boldsymbol{x}$ is already mentioned as the zero-mean unit-variance normalization. You do not need the entire dataset, unlike the standard suggestion of Python's sklearn for logistic regression which is to use a "feature-standardization" where normalization is done across all samples. These subtle and novel points are best conveyed through an impactful slideshow.
> > >
> > > EDIT: Reviewer V6Wx spotted an error in a comment in the Supplementary Material which might have led you to misinterpret what sample-standardization means. A deeply shameful, apology!

---

### Author Response · Authors · 2022-07-27
**Please keep the meta-review guideline in your mind**

Try to **counter biases you perceive in the reviews**. Unfashionable subjects should be treated fairly but often aren't, to the advantage of the papers on more mainstream approaches. To **help the NeurIPS community move faster out of local minima**, it is important to encourage risk and **recognize that new approaches** can't initially yield state-of-the-art competitive results  or **not sold according to the recipes we are used to**.

---

### Meta-Review · Area_Chair_2kVz · 2022-08-24

**Recommendation:** Reject
**Confidence:** Certain

**Metareview:**

The authors use a simple coincidence detection algorithm on pathogenic bacteria data set and increase performance by 0.5% with respect to a ResNet-26 that was published previously.

The manuscript is quite thought provoking showing that a simple algorithm can potentially outperform a complex deep neural network. However, all reviewers agreed that the study should be extended in order to be publishable at NeurIPS. In particular, they raised the following points.

- Relevance with respect to existing methods is not discussed. Related to that, the model seems similar to a perceptron with some normalization as preprocessing. This relation should be discussed.
- The method is tested only on a single data set. More evaluation would be needed in order to assess the generality of the method. Also, a clean statistical analysis is missing.
- An analysis for why method performs well on this (and potentially also other) data set is missing.

**Award:**

No

---

### Decision · Program_Chairs · 2022-09-14

Reject